# Host Status and Response Differences of Flat-Leaf and Curly-Leaf Parsley to *Meloidogyne hapla*, *M. chitwoodi*, *M. fallax*, and *M. incognita* Infestation

**DOI:** 10.3390/plants13131730

**Published:** 2024-06-22

**Authors:** Ilya Noskov, Hanna Blum, Hansjörg Komnik, Johannes Hallmann

**Affiliations:** 1Julius Kühn Institute (JKI)—Federal Research Centre for Cultivated Plants, Institute for Epidemiology and Pathogen Diagnostics, Messeweg 11/12, 38104 Braunschweig, Germany; johannes.hallmann@julius-kuehn.de; 2Institute of Crop Science and Resource Conservation—Renewable Resources, University of Bonn, Klein-Altendorf 2, 53359 Rheinbach, Germany; hblum@uni-bonn.de (H.B.); hansjoerg.komnik@t-online.de (H.K.)

**Keywords:** *Petroselinum crispum*, root-knot nematodes, aromatic plants

## Abstract

Leaf parsley growth and productivity are often affected by pathogen infection. Root-knot nematodes of the genus *Meloiogyne* are common pathogens reported on leaf parsley. The response of leaf parsley to *Meloidogyne* species in tropical and subtropical regions is quite known, while in temperate regions, comparable information is still scarce. In this study, we evaluated the host status and response of three flat-leaf (Laica, Laura, Gigante d’Italia) and three curly-leaf (Grüne Perle, Orfeo, Sombre) parsley cultivars to *Meloidogyne* species from temperate regions, i.e., *M. hapla*, *M. chitwoodi*, and *M. fallax*, as well as to the southern root-knot nematode *M. incognita*. Evaluation was based on measuring plant biomass and nematode reproduction nine weeks after nematode inoculation. Our results showed that all four *Meloidogyne* species did not cause the reduction in leaf parsley growth under the given experimental conditions. Regarding the host status of leaf parsley cultivars for *Meloidogyne*, results were variable. All six parsley cultivars were found to be good hosts for *M. hapla*. Regarding *M. chitwoodi*, the host status could not be clarified properly; however, each cultivar allowed nematode reproduction at least in one experiment. For *M. fallax*, flat-leaf parsley turned out to be less susceptible than curly-leaf parsley; and for *M. incognita*, Orfeo, Laura, and Laica were classified as good hosts, Grüne Perle and Sombre as poor hosts, and Gigante d’Italia as a non-host. Amongst all tested cultivars, Gigante d’Italia was found to be the least susceptible cultivar due to its poor host status for *M. chitwoodi* and non-host status for *M. fallax* and *M. incognita*. Infection with *M. hapla*, *M. chitwoodi*, and *M. incognita*, but not with *M. fallax*, resulted in distinct gall formation on the roots of all six leaf parsley cultivars.

## 1. Introduction

Leaf parsley (*Petroselinum crispum* (Mill.) Nyman convar. *crispum*) is an aromatic plant in the Apiaceae family that has extensive commercial applications. It is largely used worldwide as an aromatic herb for cooking and garnishing [1]. Leaf parsley is abundant in antioxidants, flavonoids, and vitamins C and A [2]. Its leaves, seeds, and roots contain bioactive compounds such as essential oils, which find broad utilization in the food industry as well as in perfumery and cosmetics [3]. There are two major types of leaf parsley, flat-leaf and curly-leaf parsley. Both types have numerous cultivars, which are primarily distinguished by their morphological traits and chemical composition [4]. Leaf parsley is grown in a wide range of climate zones, from tropical to temperate regions, and is cultivated in fields, greenhouses, and plastic tunnels [2]. In European agricultural practices, leaf parsley is usually grown as a biennial crop in rotation with field crops like potato, peas, and cereals. Its seed germination is slow, taking up to 26 days [2]. Multiple harvests per year are possible [5,6]. Leaf parsley grows well in different types of soil, including sandy soils, and its optimal growth occurs at temperatures between 7 °C and 16 °C [1,7]. Like many other members in the Apiaceae family, leaf parsley is susceptible to various pathogens [8,9], and it is not recommended to plant any other Apiaceae crop in the same field within five years after leaf parsley cultivation [10,11]. Among pathogens, root-knot nematodes of the genus *Meloidogyne* are often associated with parsley production [12,13]. It has a noticeable infection that is usually characterized by the emergence of galls on the roots of affected crops. These galls interfere with root function, impairing the plant’s ability to effectively uptake nutrients and water from the soil. The infection process starts with the infectious second-stage juveniles (J2s), hatched from the eggs, which entirely penetrate the roots where they initiate a feeding site close to the vascular tissue, called giant cells. The juveniles feed on the nutrients provided by the plant and go through the third and fourth juvenile stages until they become adult females or males. Nematode feeding disrupts plant physiology. As a result, *Meloidogyne* infection may lead to stunted plant growth and reduced yield. While there are several reports of *Meloidogyne* species affecting leaf parsley in warmer regions [14,15,16,17], little is known about *Meloidogyne* damage on leaf parsley in temperate regions. In this study, we aimed to investigate the host status and plant growth response of European flat-leaf and curly-leaf parsley cultivars towards *Meloidogyne* species from temperate regions, i.e., *M. hapla*, *M. chitwoodi*, and *M. fallax*, as well as *M. incognita*, one of the most common pathogens in subtropical and tropical regions.

## 2. Results

### 2.1. Host Status and Response of Leaf Parsley Cultivars to Meloidogyne hapla

All six leaf parsley cultivars turned out to be good hosts for *M. hapla*. According to the results, *M. hapla* reproduction rate on Grüne Perle ranged between 1.1 (Exp. 2) and 11.8 (Exp. 1), on Orfeo between 1.6 (Exp. 2) and 18.0 (Exp. 4), on Sombre between 2.5 (Exp. 2) and 7.1 (Exp. 4), on Laura between 2.8 (Exp. 2) and 29.7 (Exp. 1), on Laica between 0.6 (Exp. 2) and 8.3 (Exp. 1), and on Gigante d’Italia between 0.7 (Exp. 2) and 22.1 (Exp. 1) (Figure 1). Significant differences in nematode reproduction between leaf parsley cultivars were only found in Experiment 1, where *M. hapla* reproduction on Laura (29.7) was significantly higher than on Laica (8.3) and Sombre (7.0) (Figure 1). With this exception, flat-leaf and curly-leaf cultivars behaved similarly regarding their susceptibility to *M. hapla*. With reference to the internal control tomato, it can be said that the experimental conditions for *M. hapla* were optimal in all four trials. The average reproduction of *M. hapla* on tomato over all four experiments was 46.5 and therefore higher than in the leaf parsley cultivars.

With few exceptions, *M. hapla* densities of 2500 J2s per pot did not reduce shoot dry mass and root fresh mass of any leaf parsley cultivar in any of the four experiments. Only shoot dry mass and root fresh mass of Sombre in Experiment 4, shoot fresh mass of Laica in Experiment 2, and root fresh mass of Orfeo in Experiment 1 were significantly reduced compared to the non-inoculated control. On the other hand, *M. hapla* infection significantly increased shoot dry mass and root fresh mass in Experiment 3, with the opposite effect on the plant. Both parameters were significantly higher in inoculated plants compared to control plants (Figure 2).

### 2.2. Host Status and Response of Leaf Parsley Cultivars to Meloidogyne chitwoodi

The host status of the leaf parsley cultivars for *M. chitwoodi* could not be clearly described due to high variations between experiments. For example, reproduction of *M. chitwoodi* on Sombre was 0.9 in Experiment 5 and 0.2 in Experiment 6, but 7.2 in Experiment 7. For Gigante d’Italia, reproduction was 0.2 and 0.8 in Experiments 6 and 7, respectively, but 1.5 in Experiment 5 (Figure 3). High variations in reproduction between experiments were also shown for the other cultivars. *M. chitwoodi* reproduction on Orfeo ranged between 0.3 (Exp. 6) and 21.2 (Exp. 7), reproduction on Grüne Perle ranged between 0.3 (Exp. 6) and 5.2 (Exp. 7), on Laura between 1.8 (Exp. 5) and 9.0 (Exp. 7), and on Laica between 2.3 (Exp. 5) and 8.4 (Exp. 7) (Figure 3). In Experiment 7, reproduction on Orfeo was significantly higher than on the other five cultivars (Figure 3). Overall, *M. chitwoodi* reproduction of the six leaf parsley cultivars was lower in Experiment 6 than in Experiments 5 and 7. The average reproduction of *M. chitwoodi* on the internal control tomato was 55.9, indicating successful nematode reproduction. Despite the abovementioned exceptions, all six leaf parsley cultivars allowed *M. chitwoodi* reproduction (Pf/Pi > 1) in at least one out of the three experiments and thus leaf parsley can be considered a host for this species.

In line with the responses to *M. hapla*, the plant growth of the tested leaf parsley cultivars was in general not reduced by *M. chitwoodi*. However, there were few exceptions. In experiment 6, shoot dry mass of Laica and Laura and root fresh mass of Grüne Perle were significantly lower in *M. chitwoodi* treated plants than in non-treated plants (Figure 4). On the contrary, a significant positive effect of *M. chitwoodi* on plant growth was observed in Experiment 5 regarding shoot dry mass for all cultivars except Laica and regarding root fresh mass for the cultivars Grüne Perle, Orfeo, and Gigante d’Italia (Figure 4).

### 2.3. Host Status and Response of Leaf Parsley Cultivars to Meloidogyne fallax, M. hapla, and M. incognita

In Experiment 8, the host status of leaf parsley for *M. fallax*, *M. hapla*, and *M. incognita* varied between the six leaf parsley cultivars. Regarding *M. fallax*, Grüne Perle, Orfeo, Sombre, and Laura were classified as good hosts (Pf/Pi > 1), whereas Laica and Gigante d’Italia were considered a poor host (Pf/Pi > 0.5 and < 1) and non-host (Pf/Pi < 0.5), respectively (Figure 5). In general, *M. fallax* reproduction on curly-leaf cultivars was higher than on flat-leaf cultivars. For example, reproduction on the curly-leaf cultivar Orfeo was significantly higher than on all three flat-leaf cultivars, and reproduction on the flat-leaf cultivars Laica and Gigante d’Italia was significantly lower than on all three curly-leaf cultivars. The reproduction rate of *M. fallax* on the internal control tomato was 245.6.

The reproduction of *M. incognita* ranged between 0.0 (Gigante d’Italia) and 1.9 (Laica) (Figure 5). Differences between cultivars were not significant. Orfeo, Laura, and Laica were classified as good hosts (Pf/Pi > 1), Grüne Perle and Sombre as poor hosts (>0.5 and <1), and Gigante d’Italia as a non-host (Figure 5). The reproduction rate of *M. incognita* on tomato was 183.6. 

The reproduction of *M. hapla* on all six leaf parsley cultivars was >1, thus confirming the good host status already indicated in Experiments 1–4.

None of the *Meloidogyne* species had an impact on the plant growth of leaf parsley cultivars. No significant differences in the shoot dry mass and root fresh mass were found between control treatments and treatments with *M. fallax*, *M. hapla*, and *M. chitwoodi* (Table 1).

Considering all eight experiments, pot infestation with *M. hapla*, *M. chitwoodi*, and *M. incognita*, but not with *M. fallax*, resulted in distinct gall formation on the roots of all six leaf parsley cultivars (Figure 6). Aboveground symptoms such as wilting or stunned plant growth were not detected in any of the treatments.

## 3. Discussion

### 3.1. Effect of Meloidogyne hapla, M. chitwoodi, M. fallax, and M. incognita on Leaf Parsley

The effect of *M. hapla*, *M. chitwoodi*, *M. fallax*, and *M. incognita* on plant growth and nematode reproduction was tested in a series of eight experiments. The nematode inoculum density was 2500 J2s per pot and experiments were evaluated 8 weeks after nematode inoculation. Under those conditions, plant growth of the selected flat-leaf and curly-leaf parsley cultivars did not appear to be affected by the tested *Meloidogyne* species. This response seems to differ from that of other plants from the Apiaceae family, such as carrot and celery, where these *Meloidogyne* species are known to cause severe yield and quality damage [12,18,19,20]. In the case of the southern root-knot nematode *M. incognita*, which is known for its negative impact on leaf parsley growth [14,21], the missing effect on plant biomass could have been the result of a too low temperature or a too early evaluation. As shown by Walker [22], who also evaluated the parsley plants two months after inoculation, *M. incognita* reduced parsley dry weight at greenhouse temperatures between 20 °C and 32 °C during the day and a minimum of 16 °C at night. In the present study, temperature conditions were 20 ± 4 °C at day and night. As the total temperature was not recorded in either case, we can only speculate that the temperature in our own experiment may not have been sufficient for nematode propagation. The warm climatic conditions seem to be favorable for *Meloidogyne* infection on leaf parsley since most reports on *Meloidogyne* reducing leaf parsley growth are from the warmer regions. For example, severe losses of leaf parsley due to *Meloidogyne* have been reported for *M. incognita* and *M. enterolobii* in Venezuela [14,15], *M. arenaria* in Argentina [23] and Turkey [17], and *M. javanica* in South Italy [16]. Nonetheless, temperate *Meloidogyne* species can also reduce leaf parsley growth under favorable conditions, such as shown for *M. hapla* infecting leaf parsley roots during the winter season in the Marmara region, Turkey [24].

In contrast to the reductions in plant growth described above, some of our experiments resulted in a significant increase in shoot dry mass and root fresh mass in plants inoculated with *M. hapla* and *M. chitwoodi* in comparison to their non-inoculated controls. In those cases, plants still showed gall formation on the roots and good nematode reproduction (RF > 1), confirming suitable conditions for nematode infection and development. The fact that plant growth increased in the presence of *M. hapla* and *M. chitwoodi* is difficult to explain. One possibility could be that the plants reacted to the nematode infection with improved root growth to compensate for the damage caused by the nematode, which then also led to improved shoot growth. A further aspect could be that in reaction to nematode infection, soil microbial metabolism and nutrient availability were enhanced, resulting in improved plant growth. An example for the latter is coyote tobacco (*Nicotiana attenuate*), where *M. incognita* inoculation improved nitrogen availability in the pots that led to an increased shoot biomass [25]. Similar effects have been reported for the cyst nematode *Heterodera trifolii*, where a low amount of root infection on white clover (*Trifolium repens*) stimulated plant growth [26].

In this study, only the response of leaf parsley to *Meloidogyne* infection was investigated, while the response of another type of parsley, turnip-rooted parsley or Hamburg-rooted parsley (*Petroselinum crispum* convar. *radicosum*), was not examined. This type of parsley has an edible taproot and belongs to root vegetables, similar to another member of the Apiaceae family, carrot. Since the present study found no reduction in leaf parsley plant growth but observed root gall formation caused by *Meloidogyne* infection, which could make turnip-rooted parsley unmarketable, future research is needed to assess the damage potential of *Meloidogyne* specifically on turnip-rooted parsley.

### 3.2. Host Status of Meloidogyne hapla, M. chitwoodi, M. fallax, and M. incognita on Six Leaf Parsley Cultivars

Our knowledge on leaf parsley host status for *Meloidogyne* species from temperate regions is very limited and goes back to a few experimental studies and infestation reports. Here, we report that the host status of the six selected leaf parsley cultivars varied depending on the nematode species. For example, *M. hapla* induced galls and reproduced on all six leaf parsley cultivars. This supports findings of field surveys in Hungary [12] and Turkey [24], where leaf parsley roots were observed with galls induced by *M. hapla*. However, the *M. hapla* populations detected in these surveys were not evaluated for host status clarificationthus, no statement could be made about the host status of leaf parsley.

Regarding *M. chitwoodi*, the host status of leaf parsley could not be clearly characterized due to the high variability between experiments. Nonetheless, each cultivar was found to be a good host for *M. chitwoodi* in at least one out of the three experiments. Likewise, Brinkman et al. [20] described the curly leaf parsley cv. Moss Curled to be a good host for *M. chitwoodi*. However, in contrast to our findings, no pronounced galling was seen on the roots of ‘Moss Curled’. In the same study, Brinkman et al. [20] found ‘Moss Curled’ to be susceptible to *M. fallax*, which happened to be as for *M. chitwoodi* without pronounced galling on the roots. Similar to Brinkman et al. [20], our experiment showed a good reproduction of *M. fallax* on all tested curly-leaf parsley cultivars and no pronounced galls on the roots. Compared to the curly-leaf parsley, flat-leaf parsley cultivars seemed less susceptible to *M. fallax*, i.e., only Laura was classified as a good host, while Laica and Gigante d’Italia were classified as a poor host and non-host, respectively. These results are important for parsley growers as they give them the opportunity to grow less susceptible cultivars on fields infested with *M. fallax*.

Even though the experimental conditions were not optimal for the tropical/subtropical species *M. incognita*, root galling occurred on all tested cultivars, and at least three out of six leaf parsley cultivars, namely Orfeo, Laura, and Laica, allowed good reproduction. Leaf parsley as a good host for *M. incognita* has been reported previously by Vilela et al. [21] and Walker [22]. We assume that with a correspondingly longer experimental duration and warmer greenhouse conditions, a good reproduction of *M. incognita* on all leaf parsley cultivars would have been possible.

## 4. Materials and Methods

### 4.1. Plant Material

Six leaf parsley cultivars were selected according to their economic importance in Germany, i.e., the flat-leaf cultivars Laica, Laura, and Gigante d’Italia, and the curly-leaf cultivars Grüne Perle, Orfeo, and Sombre. Seed material was provided by breeding companies Grain Voltz (Brain sur l’Authion, France), Agri-Saaten (Bad Essen, Germany), and Enza Zaden (Dannstadt-Schauernheim, Germany), and by seed producers Kiepenkerl, (Everswinkel, Germany), and Dürr Samen (Reutlingen, Germany).

### 4.2. Nematode Material

Four *Meloidogyne* species were tested in the present study: (1) *M. hapla* Chitwood, 1949, long-term culture of unknown origin obtained from JKI Münster, Germany; (2) *M. incognita* (Kofoid & White, 1919), long-term culture of unknown origin obtained from JKI Münster, Germany; (3) *M. chitwoodi* Golden et al. 1980 originally collected in 2005 from a potato field in Borken, Germany; and (4) *M. fallax* Karssen, 1996, population 16-703, provided by ANSES, Maisons-Alfort Cedex, France, in 2021. All four populations were maintained on tomato (*Solanum lycopersicum* ‘Moneymaker’) in a sand substrate (sand + vermiculite (4:1, *v*:*v*), osmocote 3 g/kg substrate) at 20 ± 4 °C. Before the experiments, tested *Melodiogyne* populations were checked for purity following the protocol of Adam et al. [27]. Species-specific primers JMV1, JMV2, and JMVhapla were applied for *M. fallax*, *M. chitwoodi*, and *M. hapla*, and Finc and Rinc for *M. incognita*. All populations were highly virulent on tomato. To prepare a nematode inoculum for experiments, 9-week-old galled tomato roots were placed in a mist chamber and freshly hatched J2s were collected over a period of 10 days.

### 4.3. Experimental Design and Processing

Parsley seeds were sown in pots of 8 cm diameter filled with 200 mL of the sand substrate mentioned above. After three weeks, one seedling per pot was transferred into pots of 11 cm diameter filled with 420 mL of the sand substrate. One week later, plants were inoculated with 2500 *Meloidogyne* J2s in 10 mL tap water at 4 spots around the plant. Additional plants of tomato cv. Moneymaker served as the internal control for successful *Meloidogyne* penetration and development. Following inoculation, pots were organized as a randomized block design. Greenhouse conditions were 20 ± 4 °C and a 16 h light period. Plants were watered daily and insects were controlled twice a month alternating between a soft soap/ethanol solution (50 mL soft soap, 75 mL ethanol 96% in 1 L water) with the biological orange oil product PREV-AM^®^ (20 mL PREV-AM, Oro Agrl. Europe, S.A., Palmela, Portugal in 5 L water).

When egg masses of *Meloidogyne* on tomato roots were fully developed, which was 9 weeks after inoculation, plants were harvested. Nematode eggs and J2s were extracted from the plant roots using 1.5% NaOCl, as described by Hussey and Barker [28], and counted.

### 4.4. Response of Leaf Parsley Cultivars to Meloidogyne Infestation

The response of the six leaf parsley cultivars to *Meloidogyne* infestation was evaluated in a total of eight experiments. Experiments 1–4 studied the effect of *M. hapla* on leaf parsley. Non-inoculated plants served as a control. Each treatment was repeated 10 times, and each experiment consisted of 120 plants (6 leaf parsley cultivars × with/without *M. hapla* × 10 replicates). Experiments 5–7 were carried out in a similar way, but with *M. chitwoodi*. Experiment 8 consisted of six leaf parsley cultivars and four treatments (control, *M. hapla*, *M. fallax*, *M. incognita*), with five replicates per treatment.

To study the impact of *Meloidogyne* infestation on the plant growth of leaf parsley, we measured the shoot dry mass and root fresh mass. The shoot dry mass was recorded after drying the shoot material for two days at 80 °C until the weight remained constant.

Nematode reproduction (Pf/Pi) was calculated by dividing the final number (Pf) of extracted eggs and J2s with the initial inoculum number (Pi) of 2500 J2s. According to the reproduction rate, the plant host status was classified as good (Pf/Pi > 1), poor (Pf/Pi > 0.5 and <1), or a non-host (Pf/Pi < 0.5).

### 4.5. Data Analysis

Data analysis was performed using R, version 4.4.0 [29]. Differences in the growth parameters of parsley cultivars were assessed through one-way ANOVA, with graphs generated with the ggplot2 package [30]. The Wilcoxon signed-rank test was used as a post hoc for the Kruskal–Wallis Test (*p* ≤ 0.05) to compare the nematode reproduction between cultivars. Graphs were produced using the ggbetweenstats function from the ggstatsplot package [31].

## 5. Conclusions

In conclusion, our findings suggest that the six tested European leaf parsley cultivars can tolerate inoculation densities of 2500 J2s without showing any reduction in plant growth. This was the case for all four examined *Meloidogyne* species. The tested leaf parsley cultivars were susceptible to *M. hapla* and *M. chitwoodi*, and therefore should not be planted before susceptible crops such as potatoes or carrots. Flat-leaf parsley cultivars seemed to be less susceptible to *M. fallax* than curly-leaf parsley cultivars and should therefore preferably be cultivated on *M. fallax* infested fields. Also, leaf parsley was confirmed as a host for *M. incognita*. But a full coverage of host status and plant effects was not made, since the focus of this study was on the temperate *Meloidogyne* species that require more moderate growth conditions than the tropical/subtropical species *M. incognita*. Thus, any future study on the effect of *M. incognita* on leaf parsley should be performed separately under the most favorable conditions for this species.

The present work provides new and additional information on the host status of leaf parsley for temperate *Meloidogyne* species and thus will contribute to a better management of these nematodes in practical applications.

## Figures and Tables

**Figure 1 plants-13-01730-f001:**
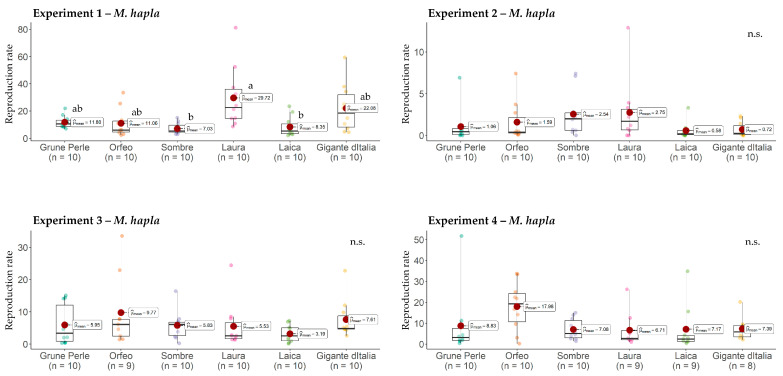
Reproduction (Pf/Pi) of *Meloidogyne hapla* on six leaf parsley cultivars. The mean reproduction rates are presented by red dots. Graphs marked with different letters on the top represent statistically significant differences between the means at *p* ≤ 0.05 according to the Kruskal–Wallis Test. n.s., non-significant.

**Figure 2 plants-13-01730-f002:**
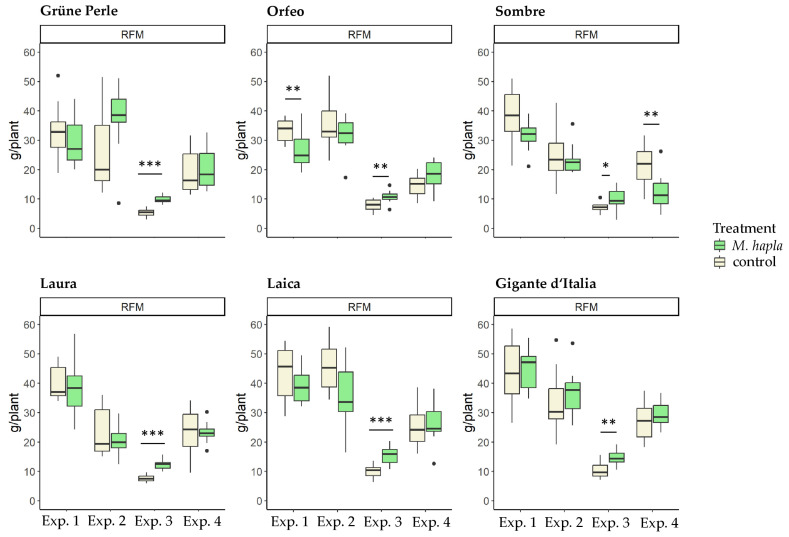
Effect of *Meloidogyne hapla* on the root fresh mass (RFM) and shoot dry mass (SDM) of curly-leaf and flat-leaf parsley cultivars. Significance of the effect of *M. hapla* on the plant parameter compared to the non-inoculated treatment represents by stars (* *p* < 0.05, ** *p* < 0.01, *** *p* < 0.001). One-way ANOVA, *n* = 10.

**Figure 3 plants-13-01730-f003:**
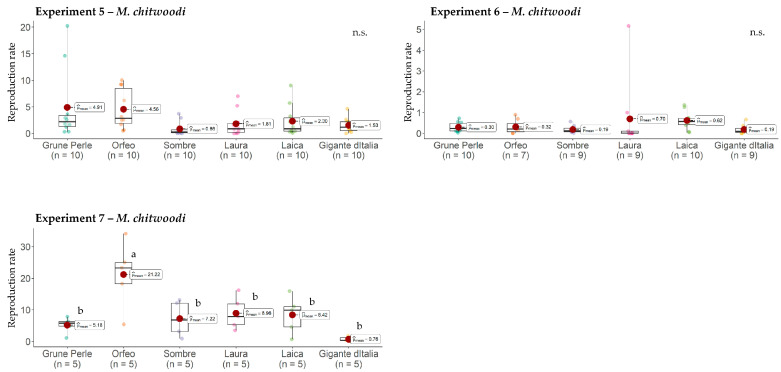
Reproduction (Pf/Pi) of *Meloidogyne chitwoodi* on six leaf parsley cultivars. The red dots represent the mean reproduction rates. Graphs marked with different letters on the top represent statistically significant differences between the means at *p* ≤ 0.05 according to the Kruskal–Wallis Test. n.s., non-significant.

**Figure 4 plants-13-01730-f004:**
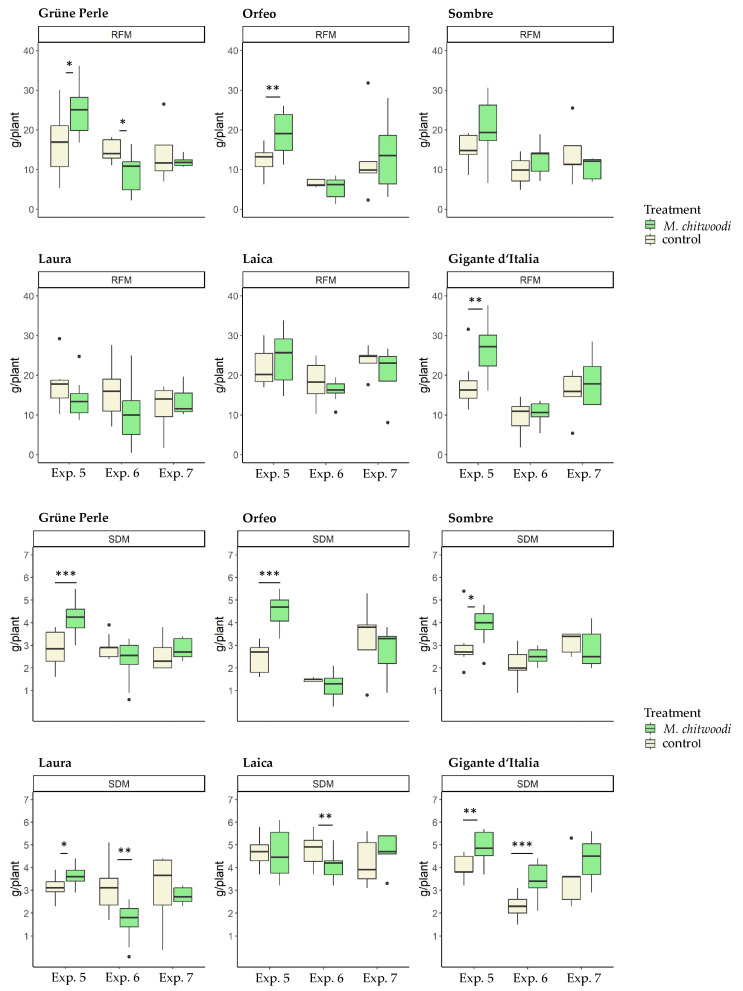
Effect of *Meloidogyne chitwoodi* on the root fresh mass (RFM) and shoot dry mass (SDM) of curly-leaf and flat-leaf parsley cultivars. Significance of the effect of *M. chitwoodi* on the plant parameter compared to the non-inoculated treatment represents by stars (* *p* < 0.05, ** *p* < 0.01, *** *p* < 0.001). One-way ANOVA, *n* = 10.

**Figure 5 plants-13-01730-f005:**
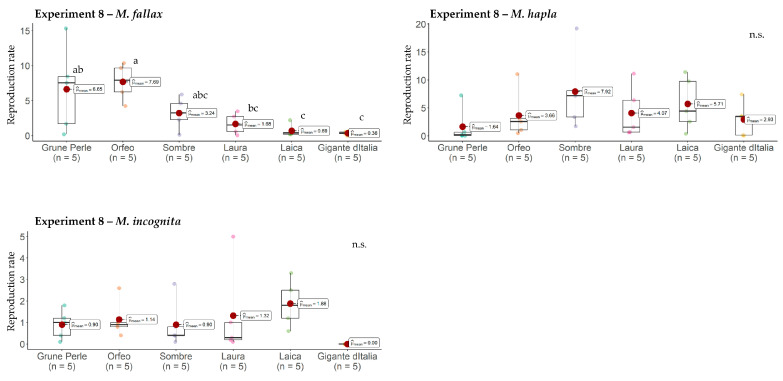
Reproduction (Pf/Pi) of *Meloidogyne fallax, M. hapla*, and *M. incognita* on six leaf parsley cultivars. The red dots represent the mean reproduction rates. Graphs marked with different letters on the top represent statistically significant differences between the means at *p* ≤ 0.05 according to the Kruskal–Wallis Test. n.s. non-significant.

**Figure 6 plants-13-01730-f006:**
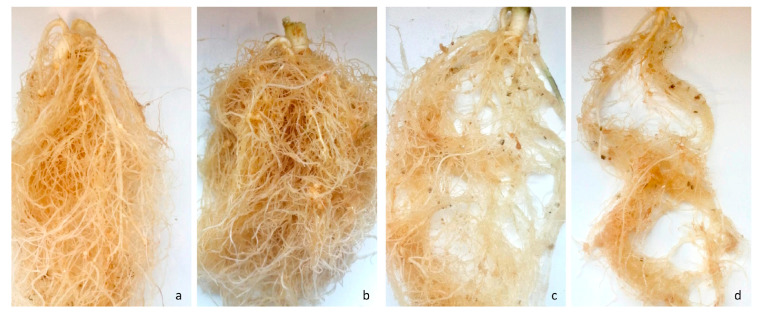
Gall formation on the roots of leaf parsley caused by *Meloidogyne hapla* on Laura (**a**) and Sombre (**b**) and by *M. chitwoodi* on Orfeo (**c**) and Grüne Perle (**d**).

**Table 1 plants-13-01730-t001:** Effect of *Meloidogyne hapla*, *M. fallax*, and *M. incognita* on the shoot dry mass and root fresh mass of leaf parsley cultivars (Experiment 8).

	Shoot Dry Mass (g)				Root Dry Mass (g)			
	Control	*Meloidogyne hapla*	*Meloidogyne fallax*	*Meloidogyne incognita*	Control	*Meloidogyne hapla*	*Meloidogyne fallax*	*Meloidogyne incognita*
Grüne Perle	2.2 ^1^	2.7	2.0	3.1	11.7 ^2^	11.3	7.5	12.6
Orfeo	3.3 ^1^	2.3	2.6	2.4	13.0 ^2^	7.6	8.3	7.7
Sombre	3.3 ^1^	2.2	3.0	2.4	14.1 ^2^	8.8	15.4	8.7
Laura	2.4 ^1^	2.9	2.8	2.3	9.4 ^2^	13.5	11.0	8.3
Laica	4.2 ^1^	5.0	4.7	3.7	23.6 ^2^	21.8	22.1	18.6
Gigante d’Italia	3.5 ^1^	5.7	3.9	3.2	15.4 ^2^	17.3	16.2	13.7

^1,2^ Means within the same row are not significantly different comparing all the treatments within one plant parameter (*n* = 5).

## Data Availability

The raw data supporting the conclusions of this article will be made available by the authors on request.

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
