# Peer review of "Host Status and Response Differences of Flat-Leaf and Curly-Leaf Parsley to Meloidogyne hapla, M. chitwoodi, M. fallax, and M. incognita Infestation"

_plants, 2024, doi:10.3390/plants13131730_

Round 1

Reviewer 1 Report

Comments and Suggestions for Authors

Dear authors,

Your study on the host status of selected parsley varieties in relation to four species of Meloidogyne is generally well-written and provides valuable insights for parsley growers, especially those with a history of Meloidogyne infestations in their fields. Consequently, I believe this work deserves publication. However, I have some questions and suggestions for improvement.

1) Please consider the most appropriate term to use: "varieties" or "cultivars" (understood as cultivated varieties).

2) Since the objective is to evaluate the host status of these vars/cvs, considering the increase in shoot dry mass and root fresh mass in plants inoculated with M. hapla and M. chitwoodi in comparison to their non-inoculated controls, it seems unnecessary and premature to conclude that parsley benefit from Meloidogyne infection.

3) There appears to be some confusion between the terms "pathogen" and "parasite" (we are discussing plant-parasitic nematodes that cause disease in plants), as well as "infestation" and "infection" (soil is infested with nematodes, but plants are infected; alternatively, one could refer to the inoculation of both soil and plants, and also say that plants are parasitised). I know the authors are familiar with these concepts, so this might have been a result of oversight or haste 😊

4) The same applies to "temperate regions" that were incorrectly referred to as "temporal regions."

4) Additionally, I suggest consistently using the same terminology for the plural of second-stage juveniles (either J2 or J2s).

Moreover, you will find minor comments and suggestions throughout the text that I would like you to review.

Best wishes for your continued work!

Rev.

Comments on the Quality of English Language

Author Response

We would like to thank editors and reviewers for commenting our manuscript. All comments have been addressed in the table below. All modifications in the text are marked in yellow (Please see the attachment)

Kind regards,

Ilya Noskov

Reviewer 2 Report

Comments and Suggestions for Authors

The paper describes biological experiments aiming research of host status of parsley with the regard to 4 species of root knot nematodes. I agree with authors that this research is quite important as there is not enough information on this topic from temperate zone and we can expect more damage caused by this important genus of plant parasitic nematodes with climate warming in future.

The quality of manuscript is quite good, the design of experiment is simple but ok for the purpose of the work. The same case are other parts of the manuscript which are presented in simple but sufficient manner.

I have only two minor remarks aiming the same issue; you should add note into Introduction on main symptom on root vegetable caused by Meloidogyne nematodes which is excessive branching of the main root which limits commercial value of the infested vegetable. Similarly, you should add into Discussion that according to your results dry and fresh mass of the infested plants do not differ from healthy ones however infested plants will be not marketable because of above mentioned symptoms. Similar results were obtained on carrots already, try to find and add citation on that into Discussion.

Author Response

We would like to thank reviewers for commenting our manuscript. All comments have been addressed in the table below. In addition, we noticed some minor inconsistences in our manuscript, which we also corrected. All modifications in the text are marked in yellow. (Please see the attachment)

Kind regards,

Ilya Noskov
